# Potential Methods of Targeting Cellular Aging Hallmarks to Reverse Osteoarthritic Phenotype of Chondrocytes

**DOI:** 10.3390/biology11070996

**Published:** 2022-06-30

**Authors:** Yuchen He, Katelyn E. Lipa, Peter G. Alexander, Karen L. Clark, Hang Lin

**Affiliations:** 1Department of Orthopaedic Surgery, University of Pittsburgh School of Medicine, Pittsburgh, PA 15217, USA; yuche@pitt.edu (Y.H.); kel159@pitt.edu (K.E.L.); pea9@pitt.edu (P.G.A.); klc99@pitt.edu (K.L.C.); 2Department of Orthopaedics, The Second Xiangya Hospital, Central South University, Changsha 410008, China; 3Department of Bioengineering, University of Pittsburgh Swanson School of Engineering, Pittsburgh, PA 15219, USA; 4McGowan Institute for Regenerative Medicine, University of Pittsburgh School of Medicine, Pittsburgh, PA 15219, USA

**Keywords:** osteoarthritis, cartilage degradation, chondrocyte, aging, senescence

## Abstract

**Simple Summary:**

Osteoarthritis (OA) is the most common chronic and disabling joint disease worldwide, causing pain and impaired mobility. Currently, no effective disease-modifying drugs are available to treat this disorder. Although not all elderly individuals have osteoarthritis, aging is considered the primary risk factor for this disease. However, the mechanisms by which aging and osteoarthritis correlate are still not completely understood. In this review, we revisit the features of OA chondrocytes and compare them with the cellular hallmarks of aging, including genomic instability, telomere attrition, epigenetic alteration, mitochondrial dysfunction, loss of proteostasis, deregulated nutrient-sensing, cellular senescence, and altered intercellular communication. It is concluded that OA chondrocytes share many similar alterations with cellular aging. Next, based on findings from studies on chondrocytes and other cell types, we propose methods that can potentially reverse the osteoarthritic phenotype of chondrocytes back to a healthier state. Lastly, we discuss the current challenges and future perspectives. Specifically, we need to identify the hub(s) that regulate(s) the most changes observed in OA and aged chondrocytes, which can subsequently assist in the development of the most efficient treatments. In addition, OA chondrocyte-targeting therapeutics may negatively influence other healthy tissues, thus warranting careful examination.

**Abstract:**

Osteoarthritis (OA) is a chronic degenerative joint disease that causes pain, physical disability, and life quality impairment. The pathophysiology of OA remains largely unclear, and currently no FDA-approved disease-modifying OA drugs (DMOADs) are available. As has been acknowledged, aging is the primary independent risk factor for OA, but the mechanisms underlying such a connection are not fully understood. In this review, we first revisit the changes in OA chondrocytes from the perspective of cellular hallmarks of aging. It is concluded that OA chondrocytes share many alterations similar to cellular aging. Next, based on the findings from studies on other cell types and diseases, we propose methods that can potentially reverse osteoarthritic phenotype of chondrocytes back to a healthier state. Lastly, current challenges and future perspectives are summarized.

## 1. Introduction

Osteoarthritis (OA) is a worldwide endemic and debilitating chronic joint disease characterized by cartilage degradation, changes in bone density, subchondral bone plate thickening, osteophyte formation, synovitis, and other changes to joint elements [1]. As the most common joint disease, OA causes many symptoms such as stiffness, joint swelling, crepitus, and functional decline, which result in reduced quality of life and increased burden on healthcare systems [2]. It has been estimated that the medical costs of OA account for about 1% to 2.5% of the US national gross domestic product and this number is expected to rise further with an aging population and the obesity epidemic [3]. In addition to physical impedance, OA also causes psychological stress, which may ultimately lead to social isolation and mental stress [4]. To date, there are no disease-modifying OA drugs (DMOADs) available to delay or reverse OA progression. Current non-surgical treatments to relieve pain mainly include anti-inflammatory drugs, opioids, and viscosupplementation (intra-articular injection of hyaluronic acid derivatives) [5]. Joint replacement surgery remains the only long-term solution for end-stage OA. However, the lifespan of prostheses is limited, and some complications have been reported [6]. Therefore, there is an unmet clinical need to develop DMOADs, whose development relies on our understanding of the etiology and pathology of OA. Through decades of studies, we have learned that OA is a heterogeneous and multifactorial disease, and some risk factors for OA have been defined, including injury, menopause, mechanical overloading/overuse, aging, obesity, specific single nucleotide polymorphisms, etc. [7]. In this review, we focus on the contribution of chondrocyte aging in OA pathogenesis.

## 2. Characterization of OA Chondrocytes

Although OA is a whole joint disease that affects all joint tissues to different extents, degradation of articular cartilage is still considered the central pathological change of OA. As has been widely acknowledged, normal articular cartilage is mainly composed of one type of cell, chondrocytes, which are responsible for maintaining the integrity of the cartilage extracellular matrix (ECM). Chondrocytes also contribute to the production of several key components in synovial fluid, such as lubricin and hyaluronic acid, that lubricate the joints [8]. When OA occurs, chondrocytes undergo multiple physiological and phenotypic changes [9]. In the early stages of injury, chondrocytes undergo phenotypic changes that increase cell proliferation and cluster formation, and enhance the production of matrix-remodeling enzymes, which together aim to repair the damage [10]. However, when self-repair mechanisms fail, OA is initiated and progresses, in which the osteoarthritic chondrocytes exhibit enlarged cell morphology, decreased proliferation ability, reduced chondrogenic commitment, and conversion to catastrophic secretory profiles [11]. Particularly, chondrocyte enlargement is a feature of OA-affected cartilage and is correlated with the severity of cartilage damage [12]. These morphologic changes observed during aging and OA share similarities with a process called chondrocyte dedifferentiation, which refers to the phenotype loss during in vitro expansion [9]. Importantly, chondrocyte dedifferentiation is associated with cytoskeleton and nuclear mechanics alteration [13,14]. The cytoskeleton is a three-dimensional (3D) network comprised of actin microfilaments, tubulin microtubules (MTs) and vimentin intermediate filaments [15]. In healthy hyaline cartilage, chondrocytes are spheroidal in shape with a small diameter and small spreading area [16]. However, in aged or degenerated cartilage, in vitro chondrocytes have an amoeboid and fibroblast-like shape with a large diameter and a large spreading area [16,17]. Moreover, a decrease in the turnover rate of cytoskeletal proteins and formation of a rigid cytoskeleton are observed in senescent chondrocytes [18]. In young chondrocytes, actin microfilaments have a cortical distribution with predominant cell periphery localization. Conversely, in aging chondrocytes, relatively fine stress fibers and large aggregates of parallel actin filaments are present [19]. A quantitative proteomic analysis in human osteoarthritic and normal chondrocytes showed that proteins belonging to pathways associated with regulation of the actin cytoskeleton were significantly enriched in the OA samples [20]. Changes in the chondrocyte cytoskeleton also regulate gene expression of catabolic factors that degrade the ECM and play a role in OA progression [21]. Additionally, osteoarthritic chondrocytes also display high levels of cellular senescence [22,23]. These alterations result in cartilage degradation and contribute to pathogenic changes to other joint elements [9].

## 3. Association between OA and Aging

In general, aging is characterized by a gradual decline in fitness with chronological age, an increased risk for disease acquisition, and an elevated proneness to death [24]. During this process, the accumulation of deleterious mutations and increased production of detrimental biomolecules, such as reactive oxygen species (ROS) and genotoxic intermediate metabolites, lead to vulnerability of tissues/cells to environmental challenges [25]. Increasing age represents the main non-modifiable risk factor for OA [26]. Specifically, the prevalence of knee OA in adults aged >60 years was 37.4% compared to a prevalence of 13.8% in adults > 26 years of age [27,28]. In another study comparing individuals between the ages of 40–49 and those > 60 years old, OA prevalence increased by approximately eight-fold [29]. Despite the strong correlation between OA and aging, the underlying mechanisms are still unclear. Increasing evidence suggests that the pathogenesis may involve both local and systemic changes. In particular, a chronic, local low-grade inflammation is involved in both the aging process and OA pathogenesis [30]. For example, a global low-grade pro-inflammatory phenotype has been observed in aged mammals, so-called “inflamm-aging” [31], which is indicated by a higher level of C-reactive protein (CRP) and cytokines, such as interleukin (IL-6) and tumor necrosis factor (TNF)-α, in the serum and the activation of nuclear factor kappa-light-chain-enhancer of activated B (NF-κB) signaling pathway [32]. Elevated levels of CRP and IL-6 were found in people with OA of the knee and levels of these pro-inflammatory markers were related to risk of progression and clinical symptoms [33,34,35]. Locally, the activation of inflammation mediators is a critical feature of OA. Joint tissue cells, including synoviocytes, chondrocytes and meniscal cells, as well as the neighboring infrapatellar fat in the knee joint, can be a local source of inflammatory mediators that increase with age and contribute to OA [36,37]. Synovitis is a frequently observed local inflammation of OA, which is indicated by synovial hyperplasia and low-grade inflammatory infiltrates within the synovial lining [38,39]. The presence of synovitis is associated with increased severity of symptoms, increased cartilage loss, decreased mobility, and elevated radiographic grades [40,41].

## 4. Cellular Hallmarks of Aging in OA Chondrocytes

After decades of research on aging, nine candidate aging hallmarks were proposed in 2013, which include genomic instability, telomere attrition, epigenetic alteration, mitochondrial dysfunction, loss of proteostasis, deregulated nutrient-sensing, cellular senescence, stem cell exhaustion, and altered intercellular communication [42,43,44]. Partially or fully ameliorating these hallmarks is believed to retard the aging process, thus increasing the healthy lifespan and suppressing the onset of aging-associated diseases [45,46]. Given that OA is an aging-associated disease and there are no efficacious DMOADs available, we propose that the current available “rejuvenation” strategies developed in the studies of other aging-associated diseases may be applied to reverse OA. In the following text, we first present an overview of current findings and examine whether OA chondrocytes also display representative aging hallmark-like changes (Figure 1). Of note, there is still a debate on whether stem cells are present in hyaline cartilage, and potential contributions to OA pathogenesis. Therefore, one of the nine aging hallmarks, stem cell exhaustion, is not discussed in this review. Next, we summarize the rejuvenation methods targeting each hallmark and the potential application in OA therapies. Lastly, the knowledge gap and future directions are discussed.

### 4.1. Telomere Attrition

Telomeres are nucleoprotein structures that cap the end of each chromosome strand and function to maintain genome stability [47]. If not elongated by telomerase, the telomere length shortens through two distinct mechanisms [48]. The first one is called replicative shortening, which occurs during each subsequent cell division. The other one is single telomere erosion, which is caused by damage to the guanine residues of telomeres [49,50]. Telomerase is a ribonucleoprotein reverse transcriptase complex, which uses its intrinsic RNA template to lengthen the telomeric G-rich strand. In most differentiated human somatic cells, the telomerase is inactive [51]. Once telomeres have shortened beyond a critical level, the cell will cease proliferation and enter either senescence or apoptosis, depending on the severity of the damage [52].

As one type of terminally differentiated cell, chondrocytes are devoid of telomerase activity [53]. Shorter telomere length was detected in chondrocytes isolated from articular cartilage of older adults as well as OA-affected cartilage, which was associated with proximity to lesions, OA severity, and senescence level [54,55]. In addition, shorter telomere length of chondrocytes is found to be associated with poorer physical performance in patients with knee OA [56], while longer telomere length may reduce the risk of hip OA [57]. Mechanistically, replication-dependent telomere shortening alone does not apply to cells with low to absent mitotic activity such as chondrocytes [50]. These non-replicative telomere erosions are probably induced by external and internal stimuli such as continuous mechanical load, abnormal inflammation, and accumulated ROS [58,59].

### 4.2. Epigenetic Alterations

Epigenetic changes during aging involve alterations in DNA methylation patterns, reduced bulk levels of the core histones, post-translational modification of histones, replacement of canonical histones with histone variants, chromatin remodeling, and aberrant production and maturation of mRNAs and non-coding RNAs (ncRNAs) [42,60]. These epigenetic alterations change the chromatin landscape, DNA accessibility, and ncRNA production in aging cells until the cells succumb to a permanent halt in cell cycle progression [61]. Enzymes involved in this process include but are not limited to: DNA methyltransferases, histone acetylases, deacetylases, methylases, and demethylases [62].

In OA progression, epigenetic changes alter the expression of specific transcription factors, autophagy activities, and the production of ECM proteins and matrix proteinases in articular cartilage [63,64]. Chondrocytes derived from OA cartilage, age-matched healthy cartilage, and even sub-groups of OA patients displayed differential methylation patterns in DNA [65]. Specifically, in OA chondrocytes, the global DNA was hypomethylated, especially regions containing matrix degradation genes (*ADAMTS4, MMP3, 9, 13*), inflammatory genes (*IL-1, 6, 8*), and OA-associated transcription factor genes (iodothyronine deiodinases (*DIO*)2, Homeobox (*HOX*) family, nuclear factor of activated T cells (*NFAT*)1, Runt-related transcription factor (*RUNX*)1, 2, *SOX9*) [66]. OA genetic susceptibility loci and SNPs have been largely superseded by hypothesis-free genome-wide association scans [67]. For example, a strong correlation between DNA methylation and transforming growth factor-β (TGF-β) expression was observed in OA patients [68]. An SNP located within the TGF-β enhancer region, rs75621460, impacted the activity of the enhancer, and showed protective effects in the cartilage and the synovium. The deacetylation of histone sidechains in OA chondrocytes was globally increased by the transcriptional co-repressors, histone deacetylases (HDAC) 1 and 2, which increased the positive charge of histones [69]. The increased net-positive histone in turn enhanced the affinity between histones and DNA, thus suppressing the expression of chondrogenic genes, such as collagen type II [66]. MicroRNA has also been shown to influence epigenetics. For example, miR-145 that suppresses *SOX9,* and miR-27a that targets *MMP13,* have an increased expression in OA chondrocytes [70]. These epigenetic alterations provide wide and novel insights into the onset, progression, and epigenetics-based therapeutic strategies for OA.

### 4.3. Mitochondrial Dysfunction

Mitochondria are the primary energy producers inside of the cell and participate in molecular biosynthesis, energy homeostasis, oxidative stress response, immune regulation, and cellular signaling transduction [71]. The importance of mitochondria in mediating cell activity and cell fate during the aging process has been well recognized [72,73,74,75]. For example, the efficacy of the respiratory chain tends to diminish with age, which consequently increases electron leakage, promotes ROS production, and reduces adenosine-triphosphate (ATP) generation [76]. Mitochondrial proteases are important multifaced regulators of mitochondrial plasticity [77]. Loss of mitoproteases is associated with aging, which severely impairs the functional integrity of mitochondria and causes many diseases [78]. Additionally, mitophagy, a special type of autophagy that removes damaged mitochondria, is also impaired in aged cells, giving rise to the progressive accumulation of defective mitochondria [79]. Compromised mitophagy and accumulated dysfunctional mitochondrial downregulate chondrocytic activity, accelerating the aging process and the development of OA [80]. In addition, aged mitochondria have decreased membrane potential and increased membrane permeability that triggers inflammatory effects via permeabilization-facilitated activation of inflammasomes [81,82,83]. Mitochondrial components, such as mtDNA and metabolites, may also leak through the highly permeable membrane into the cytoplasm and trigger the cGAS-STING or caspase-3-mediated cascade signaling pathway, eventually resulting in cell death [84,85,86].

Increased mitochondrial fragmentation, loss of mitochondrial membrane potential, diminished mitophagy activity, and accumulation of abnormally shortened or granulated mitochondria in chondrocytes have been correlated with OA progression [79,87,88]. Mitochondrial dysfunction, as well as morphology and polarization alterations, are related to the prevalence, severity, incidence, and progression of OA [89]. Mechanisms behind this connection include altered energy production, increased production of ROS and reactive nitrogen species (RNS), and enhanced inflammatory reactions [90]. For example, excessive mechanical loading impaired the mitochondrial membrane potential, resulting in increased cellular inflammation and accelerated cellular senescence [91]. Changes in the inflammatory microenvironment result in aberrant chondrocyte metabolism, shifting from oxidative phosphorylation to glycolysis. Switching between these pathways is implicated in metabolic alterations that involve mitochondrial dysfunction and play a key role in cartilage degeneration and OA progression [92]. These mitochondrial alterations also induced chronic DNA damage and increased MAPK stress signaling, both of which can act independently or together to induce senescence [22], a cell state that can cause further mitochondrial damage and the production of deleterious agents [22].

### 4.4. Loss of Proteostasis

Proteostasis, or protein homeostasis, involves the precise control of protein synthesis, folding, conformational maintenance, and the degradation of dysfunctional proteins by proteasomes or lysosomes [93]. Proteostasis prevents the accumulation of damaged components and assures the continuous renewal of intracellular proteins [42]. Collapsed proteostasis constitutes a common aging feature [94]. The mechanisms underlying the impairment or loss of proteostasis are complex, involving abnormal network connectivity amongst molecular chaperones, proteolytic machineries, and their respective regulators [95]. Dysfunctional changes in the ubiquitin–proteasomal system (UPS) and the autophagy–lysosomal system (ALS) are common causes for the impairment of proteostasis during aging. UPS and ALS are the two most important pathways for the removal of misfolded and aggregated proteins and damaged organelles [96,97].

Accumulated polyubiquitin, impaired proteasome function, and decreased expression of molecular chaperones, such as protein disulfide isomerase, calnexin, and Ero1-like protein alpha, were found in OA chondrocytes, indicating the existence and potential role of dysregulated chondrocytic proteostasis in OA progression [98,99]. Deficient prosome macropain 26S subunit non-ATPase 11, associated with reduced phosphorylated Forkhead Box O (FOX)4, was found to cause proteasomal function impairment in OA chondrocytes [98]. A study on knee cartilage tissues from young and old cynomolgus monkeys demonstrated that a loss of molecular chaperone expression correlated with aging, which resulted in the loss of proteostasis, increased ER stress and cell death [99]. Restoring proteostasis using chemical/molecular chaperones or ER stress inhibitors such as 4-phenylbutyric acid [100] and phosphorylated ERK inhibitor I [101] decreased the expression of ER stress and apoptotic markers.

### 4.5. Deregulated Nutrient-Sensing

The relationship between aging and metabolic regulation is bidirectional. On one hand, aging impairs the activity of the insulin signaling pathway, causing insulin resistance. Insulin resistance subsequently leads to hyperglycemia [102]. On the other hand, dysregulated glucose clearance capacity results in the accumulation of advanced glycation end products (AGEs), which further exacerbates metabolic dysregulation and accelerates the organismal aging process [103]. Moreover, reduced activity of serine/threonine-protein kinase STK11 and its downstream molecule 5′-AMP-activated protein kinase (AMPK) in aging cells leads to a dysfunction in energy metabolism, which is associated with reduced autophagy and age-related changes in the ECM [104].

Articular cartilage is composed of dense ECM with decreasing gradient level of oxygen and nutrients from the surface to the deep zone, which leads to the unique metabolic pattern of residential chondrocytes. Different from other somatic cells, chondrocytes largely rely on glycolysis for energy production [105]. In OA, cartilage thinning and microcracks break the oxygen tension and molecular gradients, exposing chondrocytes to an environment with relatively high oxygen tension and high level of glucose [87]. These changes convert the chondrocytes’ metabolic states to a higher aerobic metabolism ratio, increasing production of ROS byproducts [106]. An in vivo study conducted in OA murine models indicated that OA chondrocytes had increased cholesterol levels due to an upregulation of cholesterol hydroxylases (CH25H and CYP7B1) and an increased production of oxysterol metabolites [107]. In addition, obesity and OA are closely related. Excess nutrient intake, when exceeding the storage capacity of adipose tissue, leads to systemic lipo-toxicity and influences inflammatory responses [108]. The inflammatory conditions further reprogram chondrocyte metabolism towards glycolysis and lactate dehydrogenase A, subsequently promoting ROS production and inducing catabolic changes [109]. Moreover, joint homeostasis requires a complex network of growth factors to regulate the anabolic and catabolic activities of chondrocytes [110]. Growth factors including TGF-β and insulin-like growth factor (IGF-1) exhibit chondroprotective effects such as promoting chondrocyte proliferation, enhancing matrix production, and inhibiting chondrocyte apoptosis [111]. In vivo and in vitro studies conducted in human and other animal chondrocytes showed that the expression level and bioactivities of these growth factors, and their receptors decreased with aging, which resulted in deleterious effects on the structure and function of cartilage, eventually leading to OA [111,112,113]. For example, levels of TGFβ2 and TGFβ3 decrease with age as does the level of TGF-β receptors I (ALK5) and II [114]. As a result, TGFβ signaling shifts from Smad2/3 to Smad 1/5/8, leading to the expression of COL10 and MMP13 and the generation of a hypertrophic phenotype [115,116]. Interestingly, the expression of Smad3 was found to increase with age, which is probably a compensatory mechanism to maintain cartilaginous phenotype [114,117]. Additionally, a study on chondrocytes isolated from tissue donors ranging in age from 24 years to 81 years revealed an age-related decline in proteoglycan synthesis stimulated by insulin-like growth factor 1 (IGF-1) [118].

### 4.6. Genomic Instability

The integrity and stability of the genome are continuously challenged by exogenous and endogenous insults, such as physical overloading, biochemical agents, and DNA replication errors [119,120]. These challenges may lead to somatic mutations, nuclear lamina defects, deletions in mitochondrial DNA (mtDNA), chromosomal aneuploidies, and copy-number variations in cells [62,121]. In addition, deficiencies in DNA repair mechanisms exacerbate the accumulation of genetic damage throughout life [122].

Currently, the studies on genomic instability in OA chondrocytes are limited. DNA mismatch repair (MMR) is the most important post-replicative correction pathway in maintaining genomic stability. In OA chondrocytes, the levels of MMR enzymes were decreased at both the mRNA and protein level [123]. Moreover, the presence of excessive oxidative stress, a feature of osteoarthritic chondrocytes, escalates the telomere genomic instability [124]. Long interspersed nuclear element 1 (LINE-1) is a transposable element. Hypomethylation of LINE-1 in blood leukocytes induced by 8-hydroxy-2′-deoxyguanosine (8-OHdG) was associated with increased risk and radiographic severity of knee OA [125]. Increased synovial fluid 8-OHdG levels and LINE-1 hypomethylation could emerge as biomarkers indicating the severity of knee OA and may possibly take part in the pathological process of knee OA [126,127]. Interestingly, stromal cells derived from the infrapatellar fat pad (IFPSC) of OA patients maintained microsatellite stability and a sustained expression of MMR genes even at advanced culture times [128].

### 4.7. Cellular Senescence

Cellular senescence, first described by Leonard Hayflick and Paul Moorhead in the 1960s, is defined as a permanent state of cell cycle arrest [129]. The cell cycle is a complex and strictly controlled process that is mainly promoted by cyclin-CDK (cyclin-dependent kinase) complex. Under external and internal stresses, the cell cycle is suppressed at different checkpoints by different cyclin-dependent kinase inhibitors (CDKIs) [130]. Generally, cellular senescence is subdivided into two main types: replicative senescence (RS), caused by telomere loss after too many cell divisions; and stress-induced premature senescence (SIPS) induced by various damaging stimuli [44]. Based on different stimulus types, SIPS is further divided into DNA damage-induced senescence [52], oncogene-induced senescence [131], oxidative stress-induced senescence [132], chemotherapy-induced senescence [133], mitochondrial dysfunction-associated senescence [87,134], epigenetically induced senescence [133], and paracrine senescence [135]. During the aging process, chondrocytes may go through both RS and SIPS as a response to various endogenous and exogenous stresses [136]. For example, mechanical overloading, chemical toxicants, injuries, organelle stress, telomere dysfunction, permanent DNA damage, and accumulation of oxidative products are all contributors to chondrocyte senescence [137]. Senescent biomarkers such as senescence-associated-β-galactosidase (SA-β-gal), cyclin-dependent kinase inhibitors (e.g., p21 and p16), unclear senescence-associated heterochromatin foci, and components of senescence-associated secretory phenotypes (SASPs) are typically used to detect senescent cells in cultured cells and in fresh tissue samples [136,137].

OA progression is accompanied by an accumulation of senescent cells in joint tissues, especially chondrocytes in the cartilage [22]. Studies have found that the proportion and distribution of senescent chondrocytes were positively correlated with the degree of articular lesions and the severity of OA [138]. Senescent chondrocytes, especially those located in OA cartilage, exhibit distinct morphological and phenotypic alterations when compared to normal counterparts, such as an enlarged and irregular cell body, vacuolization and accumulation of stress granules, cell cycle arrest, apoptosis resistance, CDK inhibition, chromatin remodeling, and metabolic reprogramming. In addition, senescent cells display SASPs, producing cytokines (IL-1, 6, 8), chemokines (C-C motif ligand 2, CCL2) and metalloproteinases (MMP-1, 3, 12, 13) [44,131,139,140], which exacerbate inflammation and promote the degradation of ECM. In addition, these detrimental molecules might also induce senescent changes in neighboring chondrocytes through intercellular communication, further promoting OA progression [141].

Of note, recent studies have shown that the senescent phenotype in OA chondrocytes could be partially reversed. For example, our recent work indicated that restoring estrogen receptor-α level reduced p16 expression and increased the proliferation potential of OA chondrocytes [142]. Metformin was found to effectively alleviate cartilage degradation and chondrocyte senescence by enhancing the polarization of AMPK and inhibiting mTORC1 [143]. miR-140 was shown to inhibit the expression of SA-β-Gal, p16^INK^[4]^a^, and p21, effectively retarding chondrocyte senescence and attenuating the progression of OA [144]. Therefore, we speculate that OA chondrocytes are not senescent cells in the formal definition as they can proliferate in vitro. They are more similar to quiescent cells resting in the G0 stage, possessing an impaired chondrogenic phenotype due to the stresses coming from their microenvironment [11,145]. To that end, once the OA-relevant stresses are removed or reduced, these OA chondrocytes could then escape from their senescence-like state and revert back to a healthier state.

### 4.8. Altered Intercellular Communication

Aging is not solely a simple superposition of exclusive cell-autonomous alterations. Instead, it is affected by complex regulating networks and intercellular communication is highly involved. As aforementioned, senescent cells can negatively influence healthy neighboring cells via gap junction-mediated cell–cell contact, called secondary or paracrine senescence, as well as via secretion of SASP factors [146,147]. Another emerging cell–cell communication mediator in OA are exosomes. Exosomes are endosome-derived membrane-bound vesicles with a diameter ranging between 30–150 nm [148]. Informational molecules contained in exosome cargoes, such as proteins, lipids, and various DNA and RNA subtypes, change with cell types and cell conditions [148]. In OA-related research, exosomal contents were found to change with and affect OA progression [149]. For example, primary chondrocyte-derived exosomes prevented OA via restoring mitochondrial function and macrophage polarization toward the M2 phenotype [150]. However, exosomes derived from osteoarthritic chondrocytes promoted OA progression by stimulating inflammasome activation and upregulating mature IL-1β production in synovial macrophages [151]. Bone marrow mesenchymal stem cell (BM-MSC)-derived exosomes were found to protect cartilage degradation by promoting proliferation and inhibiting apoptosis of chondrocytes via miR-206/GIT1 axis [152]. Exosome-related research provides new mechanisms for understanding the pathogenesis of OA as well as novel therapeutic targets.

## 5. Potential of Targeting Aging Hallmarks to Reverse OA Chondrocytes

Potential strategies targeting different hallmarks are listed in Appendix A.

### 5.1. Increase of Genomic Stability

Eliminating oxygen free radicals and restoring the antioxidative capacity of chondrocytes are representative strategies to maintain genomic stability. Supplementation of antioxidative agents, such as ascorbic acid, nicotinamide adenine dinucleotide (NAD^+^), and activation of antioxidative molecules, such as superoxide dismutase 2 (SOD2), peroxiredoxins, and nuclear factor-erythroid 2-related factor (Nrf2), displayed protective effects and promoted cartilage formation ability [124,153,154]. For instance, NAD^+^ is an essential co-factor for DNA repair, mitochondrial function, and redox reactions [155]. Cellular NAD^+^ level declines with age, and elevating NAD^+^ was shown to inhibit DNA damage, increase DNA repair activity, improve mitochondrial function, and increase the lifespan of organisms ranging from yeast to humans [156,157,158]. Several methods have been proposed to elevate NAD^+^ levels: (1) providing exogenous NAD^+^ precursors, such as nicotinamide, nicotinamide riboside, and nicotinamide mononucleotide; (2) stimulating nicotinamide phosphoribosyl transferase to synthesize more NAD^+^; (3) inhibiting NAD^+^ degradation [159,160]. Mechanistically, NAD+ supplementation enhanced SIRT1 activity, which subsequently deacetylated and inactivated p53, as well as restored mitophagy through activating the NAD+-SIRT1-PGC1α-UCP2 axis [161,162]. Several other molecules are also found to maintain genomic stability in OA chondrocytes. For example, the zinc finger protein 16 regulates replication fork stability by recruiting the ATR/ATRIP (ataxia-telangiectasia mutated and rad3-related/ATR-interacting protein) complex [163]. Wogonin, a natural flavonoid, intercalated with genomic DNA and caused a reduced fragmentation of genomic DNA via suppression of IL-1β-mediated ROS induction [164]. Elevating the inhibitor of NF-κB kinase to ameliorate the function of the DNA MMR system was shown to help maintain genomic stability [123]. However, if deployed in an inappropriate cellular context, such as improper cell cycle phase, these same repair functions can mediate chromosome rearrangements that underlie various human diseases, ranging from developmental disorders to cancer [165].

### 5.2. Elongation of Telomeres

As mentioned above, telomerase is capable of extending telomere length. Strategies proposed to reconstruct or enhance enzymatic activity include: (1) gene therapy with transfection of the telomerase gene; (2) re-expression of silenced telomerase in somatic cells; (3) activation of residual enzymatic activity; (4) modulation of the intracellular location [166] (Appendix A). Transient ectopic expression of the catalytic subunit of telomerase, telomerase reverse transcriptase, using adeno-associated virus serotype 9-based gene therapy in adult mice increased both health span and life span without increasing cancer incidence [167]. Cycloastragenol (commercially available as TA-65) is a telomerase activator derived from the *Astragalus membranaceus* root [168]. It enhanced telomerase expression via activating ERK (extracellular regulated protein kinases) pathway in immune cells, neonatal keratinocytes, and fibroblasts, elongating telomere length without increasing cancer incidence [169,170,171]. Resveratrol was found to activate telomerase via upregulating SIRT1 [172]. Another compound (AGS-499) activated telomerase and displayed neuroprotective effects in SOD1 transgenic mice [173]. Indirect strategies to upregulate telomerase activity were achieved by supplementing metabolites, such as N-acetylcysteine, α-tocopherol, HMG-CoA reductase inhibitors and Ginkgo biloba. These metabolites either block the nuclear export of telomerase into the cytosol [174], retain telomerase activity [175], or activate telomerase via induction of Wnt/β-catenin or PI3K/Akt signaling pathways [176,177].

At present, telomere elongation or telomerase activation strategies have not been applied in treating OA. This is probably due to the avascular nature and dense matrix structure of cartilage that prevents the interventions. For instance, the dense cartilage ECM makes it difficult to transfer drugs and vectors into chondrocytes at effective concentrations. In addition, unlike stem cells and other cells that can be obtained through a relatively easy procedure without causing significant tissue lesions, chondrocytes can only be extracted from cartilage, which unavoidably damages cartilage. Moreover, there are safety concerns regarding the use of telomerase. For example, uncontrolled induction of telomerase has the potential to induce tumorigenesis [178]. Moreover, it has also been shown that extreme telomere length does not provide benefits to cellular fitness, aging, or senescence in yeast cells [179].

### 5.3. Epigenetic Modifications

The reversible nature of epigenetic mechanisms makes it possible to restore or reverse aging-related changes [180] (Appendix A). Sirtuins are a family of highly conserved histone deacetylases that regulate histone deacetylation and lifespan extension, whose levels decrease with aging [181]. Increased expression levels of Sirtuins, especially SIRT2 and its homologues, extended the lifespan of budding yeast *S. cereviseasae*, worms *C. elegans*, fruit flies *D. melanogaster*, and mice [181,182]. Sirtuins achieve prolongevity effects by interacting with several lifespan regulating signaling pathways, such as insulin/IGF-1 signaling pathway, target of rapamycin (TOR), AMP-activated protein kinase (AMPK), and forkhead box O [182,183]. High-throughput screening has identified over 14,000 Sirtuin-activating compounds, providing a huge collection of candidates for future investigation [182]. Another rejuvenation strategy through epigenetic modification is caloric restriction (CR). This influences epigenetic processes via regulating DNA methylation [184], RNA splicing [185] and histone modification [186].

Current gene therapy for OA primarily focuses on the overexpression of therapeutic factors such as growth or transcription factors, or the suppression of genes that cause joint destruction [187]. In most applications, viral or nonviral vectors were used as a vehicle to deliver gene-based therapeutic agents into the joint space [188]. However, the low efficacy of nonviral vectors, and risk of insertional mutagenesis events provoked by lentiviral vectors, limited their applications [189]. Instead, nonpathogenic, replication-defective human recombinant adeno-associated viral (rAAV) vectors have attracted much attention, which do not integrate into the genome [190]. In addition, rAAV showed very high gene transfer efficiencies within the dense ECM [187]. Recent studies showed that hydrogel-guided rAAV-mediated IGF-I overexpression provided long-term cartilage repair and protection against perifocal osteoarthritis in a minipig full-thickness chondral defect model [191]. The rAAV-mediated overexpression of *SOX9* gene enhanced osteochondral repair in sheep [192] and protected human articular chondrocytes against deleterious effects caused by osteoarthritis-associated inflammatory cytokines [193]. Polymeric micelles are effective rAAV controlled delivery systems. Human studies showed that rAAV-mediated overexpression of *TGF-β* in polymeric micelles stimulated the biological and reparative activities of human articular chondrocytes in vitro and in a human osteochondral defect explants model [194].

Studies on epigenetic modulations in chondrocytes focus on different regulators. The best-known example is miR-140. Transfection of miR-140 showed protective effects by suppressing the expression of ADAMTS 4 and 5, negatively regulating p38 mitogen-activated protein kinase (MAPK) signaling and acting as a negative feedback regulator of MMP13 [195]. Other examples include miRNA-222, -122 and -146a. miRNA-222 has been found to target histone deacetylases 4 [196], while miRNA-122 targets deacetylase SIRT1, to control histone modifications and ECM degradation [197]. Histone deacetylase inhibitors, SAHA (vorinostat) and LBH589 (panobinostat) negatively regulated IL-1β signaling through elevating miRNA-146a expression [198]. Further epigenetic studies are required to determine the upstream regulators that trigger these epigenetic events and the interactions among the different epigenetic mechanisms.

### 5.4. Restoration of Mitochondrial Function

Research on restoring mitochondrial function showed promising results against aging related changes. For example, decline in oocyte quality and quantity caused by impaired mitochondrial performance could be reversed by maternal dietary administration of coenzyme Q10 (CoQ10) [199]. Restoring mitochondrial DNA copy number was shown to preserve mitochondrial function and delayed vascular aging in mice [200] (Appendix A). Spermidine (SPD) is a major mammalian polyamine, which was shown to increase the expression of SIRT1, PGC-1α, Nrf1, Nrf2, and mitochondrial transcription factor A, decrease ROS production, and improve oxidative phosphorylation performance in senescent cardiomyocytes [201]. In addition to repairing or preventing mitochondrial damage, there are other methods to restore mitochondrial function, such as promoting mitochondrial biogenesis, stimulating the degradation of damaged mitochondria, or co-opting mitochondrial function to induce cell death [202].

Several methods have been proposed targeting mitochondrial function for OA treatment. Trehalose is a natural disaccharide that presents in a diverse range of organisms. It exerts cell-protective effects under various stress conditions by inducing autophagy via mTOR-independent pathways [203,204]. Trehalose protects mitochondria in chondrocytes by ameliorating oxidative stress-mediated mitochondrial membrane potential collapse, promoting dynamin-related protein 1 translocation into the mitochondria, and upregulating proteins in the mitochondria and ER stress-related apoptosis pathway [205]. Sirtuin 3 is an NAD+-dependent histone deacetylase mainly located in mitochondria. It regulates mitochondrial function, regeneration, and dynamics in order to maintain redox homeostasis and prevent oxidative stress in cell metabolism [206]. Pharmacological activation of AMPK-SIRT3 signaling by A-769662 showed chondroprotective effects by preserving mitochondrial DNA integrity and function [207]. CGS21680 is a kind of adenosine A2A receptor agonist. In vivo intraarticular injection of CGS21680 in OA mice joints and in vitro addition to the culturing medium of human chondrocytes showed protective effects on mitochondrial metabolism and mitigated ROS-mediated mitochondrial injury [208]. The prolylhydroxylase inhibitor, dimethyloxalylglycine, was found to alleviate mitochondrial dysfunction and apoptosis under hypoxia stimulation [209]. Blocking JNK/cFos-AP1 pathway with cFos/cJun inhibitor (T5224) suppressed the mitochondrial dysfunction-induced expression of catabolic genes in chondrocytes [210].

### 5.5. Improvement of Proteostasis

Interventions that improve protein homeostasis are limited, with few studies available at present (Appendix A). Interventions that specifically improve protein homeostasis in chondrocytes are even more scarce. Here, we provided the latest studies on neurotrophin signaling, aging of the brain and heat shock to inspire similar studies on chondrocyte aging and OA. These methods include but are not limited to: overexpression of mitochondrial-targeted catalase, suppression of IGF-1 signaling, and supplementation of rapamycin for alleviating defects in proteostasis in aging models [211,212]. Promoting EFK-1 signaling and increasing heat shock factor (HSF)-1 binding are other effective methods for lifespan extension and aging deceleration [213,214]. Mitophagy is another way for cells to degrade aberrant components when stress-induced damage within mitochondria exceeds the maximum processing capacity of mtUPR [215,216]. Autophagy receptor p62/sequestosome (SQST)-1 facilitates the degradation of ubiquitinated cargo and promotes proteostasis and longevity in C. elegans by autophagy induction [217].

### 5.6. Metabolism Interventions

Given its noninvasive and convenient nature, dietary intervention represents a promising strategy to restore healthy metabolism (Appendix A). The most well-defined dietary intervention to delay aging is calorie restriction (CR), which was first proposed in the 1930s, defined as a reduction of calorie intake to a level that does not compromise overall health [218]. CR has been identified as a non-invasive metabolic manipulation method that can be applied to lessen OA [219]. CR not only reduces weight-bearing burden on joints, but also improves systemic inflammatory status via preventing aging-associated global changes in DNA methylation, histone modification and chromatin remodeling [220]. Considering that continuous reduction of calorie or food intake is not easy to practice in the daily lives of most individuals, fasting-related interventions such as intermittent fasting (IF) and time-restricted feeding (TRF) have emerged as alternatives of CR [221]. Employing the Mediterranean Diet, IF and TRF can mimic CR variations and have been shown to extend lifespan in animal models [221,222,223]. Naturally derived bioactive compounds found in foods, dietary supplements, and herbal products also have the potential to influence metabolism, including (1) anthocyanidins, which are rich in fruit with purple skin; (2) resveratrol, an antioxidant polyphenol contained in grape skin; (3) quercetin, a flavonoid found in fruits and vegetables; (4) flavanols that are particularly contained in cocoa products; and (5) curcumin, a hydrophobic polyphenol produced by Curcuma longa [224,225]. With regards to pharmaceutical interventions, some drugs appear promising in reducing anabolic activities and are already being used for other clinical applications, such as rapamycin (or Sirolimus) and metformin [226]. However, notable side effects need to be first addressed [227].

Genetic polymorphisms or mutations that reduce anabolic activities such as growth hormones, IGF, IIS or downstream intracellular effectors such as AKT, mTOR, and FOXO, have been linked to longevity [228,229]. Pharmacological manipulation that suppresses nutrient signaling to reduce anabolic activities and extend lifespan includes GH/IGF1 axis and PI3K inhibitors [42]. Adenoviral overexpression of CH25H or CYP7B1 in mouse joint tissues caused OA, whereas knockout or knockdown of these hydroxylases rescinded the pathogenesis of OA. Thus, the CH25H-CYP7B1-RORα axis of cholesterol metabolism may provide a therapeutic avenue for treating OA [107]. Additionally, dietary intake of strawberries and blueberries was shown to alleviate pain and inflammation in knee OA [230,231], probably via supplying antioxidants. However, further investigations are needed in this field.

### 5.7. Mitigation of Cellular Senescence

At present, whether chondrocytes in OA cartilage permanently lose their proliferation capacity is controversial. In our recent study, we found that chondrocytes from severely damaged aged cartilage exhibited senescent markers and produced classic SASPs [142]. However, these chondrocytes were still able to re-enter cell cycle and proliferate, which is contradictory to the definition of senescence [232]. Furthermore, the expression of senescent markers and impaired chondrocyte function can be at least partially reversed through regulating gene expression [142]. Therefore, we speculate that chondrocytes in damaged cartilage are not senescent cells in the formal definition. Their inferior phenotype can be reversed, thus representing an efficacious therapeutic target [21].

Inhibition of p38, disruption of p53 and p16, and extension of telomere length strategies have been shown to reduce senescent phenotypes but carry a major caveat, as they can potentially increase the incidence of cancer [233]. In recent years, researchers have started to explore novel approaches to target senescent cells in OA cartilage. Among them, senolytics and senostatics have attracted much attention [22]. Senolytics are a class of drugs that selectively eliminate senescent cells via inducing cell apoptosis, which has been revealed as a promising method in treating aging-related diseases in murine models, such as idiopathic pulmonary fibrosis, atherosclerosis, and cancer [234]. Several senolytics have been investigated in OA models, and promising results were reported. For example, ABT-263 (navitoclax) [235] and FOXO4 D-retro inverso isoform [236] have been shown to selectively kill senescent chondrocytes. Mechanically, senolytics are designed to induce death of senescent cells by targeting the Bcl-2 family of antiapoptotic proteins (e.g., ABT263 and ABT737), by promoting nuclear exclusion of p53 (e.g., FOXO4-DRI), by targeting glycolysis (e.g., quercetin and 2-deoxyglucose), or other pathways that lead to apoptosis (e.g., dasatinib) [237]. Other identified senolytics, including dasatinib, quercetin, fenofibrate, fisetin, curcumin, piperlongumine, ouabain, o-vanillin, panobinostat, tanespimycin, alvespimycin, and UBX0101, also possess potential as therapeutics for OA [238,239,240,241,242,243]. Of note, many of these drugs bear intrinsic toxicities and target only subsets of senescent cells without discriminating among beneficial and deleterious senescence programs, which could limit their use in clinical applications [44]. In addition, the reappearance of senescent cells after cessation of senolytic treatment remains a concern [240]. Another potential issue is that the removal of senescent cells may result in a lack of sufficient residual cells to maintain the normal structure and function of cartilage.

Senostatics are another effort towards anti-senescence. Instead of killing cells, senomorphics aim at reducing the detrimental effects of senescent cells by inhibiting NF-kB or other pathways controlling the secretory phenotype [244]. Fisetin, rapamycin, ruxolitinib, loperamide, niguldipine, apigenin, urolithin A and kaempferol all are within the senomorphic category [22,91,245].

Heterochronic parabiosis represents another practical method to reduce the number of senescent cells and the level of SASPs. The method consists of the circulatory systems of young and aged animals being surgically attached, thereby facilitating the exchange of immune cells and secreted factors present in the blood [246]. In heterochronic parabionts, the age-dependent increase in senescence and SASP marker expression is reduced in old mice exposed to a young environment [247]. The beneficial effects of young blood on aged muscle regeneration was probably achieved via extracellular vesicles (EVs) contained in the serum [248]. Another classical therapeutic approach is applying the differentiated cells from MSCs or induced pluripotent stem cells (iPSCs) to regenerate or replace cells inside a damaged tissue or even replace the entire injured organ [249].

### 5.8. Reduction of Detrimental Intercellular Communication

Several options have been proposed to restore defective intercellular communications that are lost with aging, including genetic, nutritional, and pharmacological interventions [250,251] (Appendix A). SASPs are crucial contributors to detrimental communications that not only maintain the senescent state but also induce pathological changes in healthy chondrocytes. Strategies targeting SASPs have been discussed in detail above. Another important signal mediator is the exosome. Taken up by cells via endocytosis, direct membrane fusion, and pinocytosis, exosomes regulate intercellular communications, such as affecting gene expression in targeted cells [252]. Exosomes derived from platelet-rich plasma are effective in delaying the progression of OA via promoting proliferation and inhibiting apoptosis of chondrocytes via activation of the Wnt/β-catenin signaling pathway [253]. MSC-derived exosomes prevent OA development by attenuating inflammation and restoring chondrocyte mitochondrial function [254]. Additionally, bioengineered exosomes, with modifications either on the membrane surface or intracellular contents, are promising biomolecule and drug delivery cargos for information transfer and the regulation of cell communication [255]. Another category of important molecules involved in communication regulation are the damage-associated molecular patterns (DAMPs), which accumulate with age and are released in the extracellular media [256]. DAMPs bind pathogen-recognition receptors (PRRs), such as Toll-like receptors (TLRs), and subsequently activate innate immunity and lead to joint inflammation [256]. Therefore, inhibiting the production and transduction of DAMPs or PRRs represents a potential therapeutic technique for OA [257].

Primary human chondrocytes in tissue and in monolayer culture were found to contain high levels of connexin 43 (Cx43) and were able to directly communicate through gap junction channels [258,259]. Interestingly, primary bone cells, synovial cells and chondrocytes are able to establish cellular contacts and communicate through gap-junction channels with connexin43 (Cx43) as well [260]. Andrew et al. evaluated the number of gap junctions, the amount of gap-junction proteins, and the amount of enzymatic activity mediated by gap-junctions in synovial lining cells. They found that gap-junctions between synovial lining cells increased significantly in patients with OA, accompanied with higher expression levels of matrix metalloproteinases in response to IL-1β stimulation [261]. Overexpressing Cx43 in cultured rabbit and human synovial fibroblast cell lines enhanced the expression of osteoarthritis-associated genes, including *MMP1*, *MMP13*, *ADAMTS4*, *ADAMTS5*, *IL-1*, *IL-6* and *PTGS2*, and increased the secretion of collagenases [262]. Similar changes were also observed in chondrocytes. Significantly elevated levels of Cx43 and Cx45 were found in chondrocytes from OA patients compared with chondrocytes from normal cartilage [263]. Overactive Cx43 in OA chondrocytes increases the expression of tissue remodeling enzymes and proinflammatory agents, and regulates proteins related to nucleolar functions, RNA transport, and translation [264,265]. These studies implied the therapeutic potential of targeting gap junctions to interact detrimental intercellular communication in OA. A recent study using human chondrocytes found that downregulation of Cx43 either by CRISPR/Cas9 or carbenoxolone treatment triggered redifferentiation of OA chondrocytes, decreased synthesis of MMPs and proinflammatory factors, and attenuated cellular senescence [264]. Of note, although the modulation of connexins and pannexins represents an appealing therapeutic target in joint disease, they are important in maintaining normal cellular interactions. Their complex regulation, their combination of gap-junction-dependent and -independent functions, and their interplay between gap-junction and hemichannel formation also require further research.

## 6. Healthy Aging of Chondrocytes

In the content above, we focused on OA-accompanied aging. In the aging population, a large portion of individuals are not affected by OA. Due to the challenge of collecting chondrocytes from this OA-free cohort, we actually know very little about the changes of chondrocytes in healthy aging. This information will be critical to developing methods to prevent or reduce the onset of OA, since aging is thought to increase the susceptibility of chondrocytes to other OA-inducing agents, such as oxidative stress and inflammation [104,266,267]. We can target these healthy aging-associated changes to reduce the onset of OA. Based on current findings, during the healthy aging process, changes in chondrocytes include but are not limited to altered metabolic activity, decreased sensitivity to growth factors, decreased proliferation rate, flattened cell shape, hypertrophic tendency, cellular senescence, telomere changes, and impaired DNA repair mechanisms [21,268,269]. For example, we recently found that chondrocytes isolated from healthy aged donors showed reduced proliferation potential when compared to cells isolated from healthy young donors, which was associated with increased p21 expression [21]. Moreover, old chondrocytes displayed impaired capacity in generating hyaline cartilage, indicated by the high level of chondrocyte hypertrophy and cellular senescence in the tissues created by these cells. In fact, hypertrophy and senescence are also two representative pathological features of OA [270].

## 7. Conclusions and Future Perspectives

In this review, we present extensive evidence supporting that the osteoarthritic conversion of chondrocytes is also an aging-like process. However, some specific aging-relevant changes, including genomic stability and intercellular communication, need to be further studied in OA chondrocytes. In addition, we summarize the “anti-aging” agents/methods that demonstrate potential in treating OA, which can be validated in animal studies in the future.

For a long time, aging has been considered as the most important risk factor of OA, and has therefore gained much attention. Compared to other published reviews on the related topic [105,136,271,272,273] our work here systematically discusses the pathologic roles of all nine aging hallmarks as they relate to OA onset and progression. Additionally, we summarize the available updates on the experimental and potential clinical approaches for treating OA through targeting the proposed aging hallmarks.

As outlined above, OA chondrocytes display multifaceted changes, including those that are similar to hallmarks of aging. Although single molecules/pathways all represent potential therapeutic targets, the most efficient method is to identify the most upstream key alteration. However, there is currently a knowledge gap in connecting the risk factors with the first molecular responder(s) that lead to irreversible OA progression. In fact, some current aging-suppressing methods simultaneously influence several hallmarks, suggesting they may target a common mechanism. For example, upregulating NAD^+^/NAD ratio via nutrient therapy displays mitochondrial protective effects, decreases ROS production, and alleviates oxidative stress, which subsequently protects cells from genome instability, senescence, protein malfunction, and inflammation. Through systemically studying changes of all aging hallmarks, we may be able to identify the hub(s) that regulate(s) all these observed changes in OA chondrocytes, thus defining more efficient treatments. Exercise for non-surgical management of osteoarthritis has been proven effective in reducing pain, improving function and performance in people with knee and hip OA [274,275]. Moreover, based on in vitro studies, exercise is also an important intervention strategy to delay aging and the retardation of chondrocytes. For example, studies found that cyclic compression at frequencies of 0.01–1 Hz, with strain amplitudes of 1–5%, was beneficial to in vitro chondrocyte proliferation and chondrogenesis by inducing the release of growth factors or transiting latent transforming growth factors (TGF) into their active forms [87,91].

In addition, it is possible that intervention on one hallmark may deteriorate other hallmarks. Therefore, a global examination of cell phenotype after treatment in preclinical models is critical to avoid side effects in future clinical trials. Moreover, it is known that cartilage degradation displays as heterogeneous during OA progression. Well-preserved and severely damaged areas are observed in the same OA knee joint [142]. Therefore, the application of treatments to target cells in the damaged area should not negatively influence the cells in the intact area. For example, in one of our in vitro studies, we found that senolytics dasatinib plus quercetin improved the phenotype of chondrocytes from damaged cartilage. However, they caused a detrimental impact on the chondrogenic potential of chondrocytes from intact cartilage [276].

Another area in which advancements are required is the delivery of therapeutic agents in a safe and efficient manner. Cartilage ECM has a dense structure with a high-density negative charge, which impedes the infiltration of different types of drugs to reach an effective concentration in chondrocytes. In addition, in situations where the therapeutics are delivered intraarticularly, repeated administration is practically difficult. Therefore, a delivery system that can release drugs in a controlled manner but still display robust cartilage-penetrating capacity is desired. As for gene therapy, genetic material can be effectively delivered into cells by nonintegrating viral vectors, nonviral vectors as well as exosomes. At present, different viral vectors, such as adenoviral, adeno-associated viral, integration-deficient retro-lentiviral, poxviral vectors, and nonviral vectors, such as plasmid vectors and artificial chromosomes, have been applied in preclinical and clinical studies [277,278,279].

Lastly, although chondrocytes are the focus of this review, it should be noted that OA represents a whole joint disease. Tissue–tissue crosstalk has been shown to play a critical role in OA progression. As discussed above, alterations in cellular communication are observed in OA. The safety and efficacy of treatments that are developed to target chondrocytes need to be tested on other joint elements as well. Recently, a microphysiological knee joint system was developed [280], which can serve as a high-throughput platform to assess the influence of treatments on all joint elements simultaneously.

## Figures and Tables

**Figure 1 biology-11-00996-f001:**
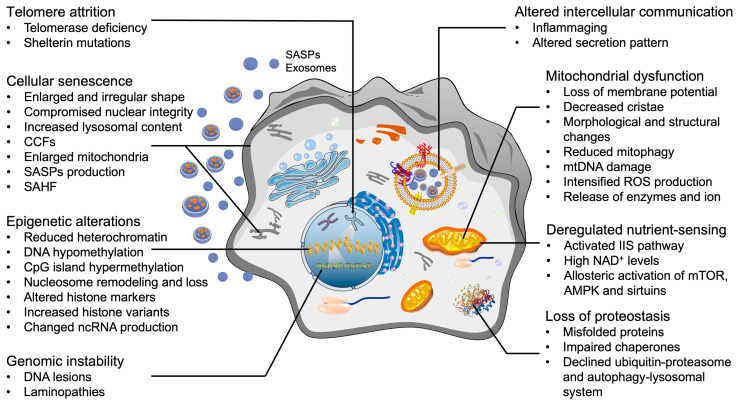
**Cellular hallmarks of aging that are also present in OA chondrocytes.** SASPs: senescence-associated secretory phenotypes; ROS: reactive oxygen species; CCF: cytoplasmic chromatin fragments; SAHF: senescence-associated heterochromatin foci; mTOR: mammalian target of rapamycin; AMPK: AMP-activated protein kinase; NAD: nicotinamide adenine dinucleotide.

## Data Availability

Not applicable.

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
