# Peer review of "Potential Methods of Targeting Cellular Aging Hallmarks to Reverse Osteoarthritic Phenotype of Chondrocytes"

_biology, 2022, doi:10.3390/biology11070996_

Round 1
Reviewer 1 Report
This review pointed out eight cellular hallmarks of aging that could play potential roles in the pathophysiology of OA. For each, the authors discussed how these hallmarks affect the cellular senescence and gave suggestions about the potential methods to reverse or alleviate the ageing process of chondrocyte. At last, the future perspectives was noted, Compared to other existing reviews on OA and chondrocyte aging, this is a moderate review. In general, this article is more like an accumulation of evidence, rather than a fluent and logical story. Therefore, this article is still a long way from publication and needs major revision.
Major comments
- In OA, aging is a more complex physiological process compared to the dedifferentiation of chondrocyte. The authors describe eight hallmarks, but unfortunately, each of them seems to be not described thoroughly and in detail, and some examples provided seem not to not focused on aging and not address the topic in the corresponding section (e.g. Line 443-453). The authors should go through the text and change the examples and / or the references accordingly. An enriched comparison and discussion is also necessary in order to distinguish the different effects of aging and dedifferentiation of chondrocyte on OA.
- Line 280: After reading the reference 101 (Ramasamy, T. S. Chondrocyte aging: The molecular determinants and therapeutic opportunities) here, I found that the current review has much of the same content as that one. Therefore, the novelty of this article is poor. Please novelty of the present work needs to be strongly highlighted compared to this and / or even other references.
Minor comments
- The title is not appropriate. As the authors state, the proposed methods for potentially reversing OA chondrocytes is mostly based on the findings from other cell types and diseases. Little evidence is based on data from chondrocytes and OA. Thus, the title needs to indicate these speculative parts.
- Line 57: The authors state that “….OA is a combination of local and systemic changes”. But the following examples in lines 58 to 63 talks about inflammation in general. How inflammation affects local and systemic changes is not mentioned or described.
- Line 64 and line 46: If section 1 and 2 were sequentially interchanged, it would make the structure more logical. It is better to introduce OA and then the association between OA and aging.
- Line 77: the changes of cytoskeleton is vital importance in the enlargement of the chondrocyte, which should be mentioned and discussed here.
- Line 92: This sentence has lots of information and more references are suggested to provide proof of the nine candidates, i.e., that these are the most important 9 aging hallmarks.
- Line 164 and 170: The authors stated that the importance of mitochondria in mediating cell activity has been well recognized. But they only gave two examples here and concluded the changes cause mitochondrial deterioration and global cellular dysfunction, which is cursory. More details should be provided here.
- Line 244, the authors state the the role of growth factors in OA has been widely reported. How do growth factors influence cell aging?
- Line 263-265: How it is contradictory to which findings? Why?
- Line 271: it is better to put this sentence first, and then talking about the cell cycle.
- Line 309-312 and 493-495: The authors pointed out his/her own inferential conclusion. However, the evidence to support these conclusions are not provided.
- In the section 4 Potential of targeting aging hallmarks to reverse OA chondrocytes, the authors focus on SIRT1/2/3 (i.e., in Sections 4.1, 4.2, 4.3, and 4.4). It is well known that members of Sirtuin family play a role in cell function. The authors need to provide other evidence here and to avoid the use focusing on one particular one particular mechanism (or pathway or family) to explain everything, because it is not a review about Sirtuins.
- Line 443-453: Again, the text here has nothing to do with either OA or chondrocyte aging. The citations the authors refer to are about neurotrophin signaling, aging of the brain, heat shock? Please explain a bit more and add appropriate citations related to OA and chondrocyte aging.
- Line 543 Section 4.8: when talking about intercellular communication, gap junctions are necessary to be discussed.
- Line 542-544: “Exosomes derived from platelet-rich plasma alleviated knee OA by promoting proliferation and inhibiting apoptosis of chondrocyte via Wnt/β-catenin signaling”. Is it related to intercellular communication?
Reviewer 2 Report
In this review, authors discuss aging as a possible contributing factor in chondrocyte dysfunction and subsequent osteoarthritis, and possible interventions for aging-related OA pathogenesis.
- Several reviews extensively focus on the 'nine hallmarks of aging'. The current review would benefit from a focused discussion of the studies that demonstrate association between the aging-associated pathways in chondrocytes and OA progression.
- For example mitochondrial dysfunction leads to generation of ROS in chondrocytes. Likewise, deficiency of prosome macropain 26S subunit non-ATPase 11 leads to impairment of proteasome function. Do chondrocytes undergo apoptosis resulting in accelerated degeneration of the cartilage?
- It would also be beneficial to discuss at what stage of disease progression are some of the aging-associated pathways identified in OA. In several instances, authors refer to changes in OA chondrocytes isolated from patients- are the various aging-related pathways activated as a result of end stage disease? Does activation of an aging-associated pathway in a non-diseased, young chondrocyte lead to cellular degeneration and result in OA (i.e. discuss correlation vs. causation)?
- Details and clarification is required in several instances.
- For example, lines 231-233 'Studies indicated that osteoarthritic chondrocytes...', was this an in vivo, in vitro or a clinical study?
- Which of the aging-associated pathways are predominantly associated with OA pathogenesis? It would be beneficial to provide statistical data on genes/proteins associated with one (or several) pathways that are dysregulated in OA chondrocytes
- The table highlights the various therapeutic strategies to reverse aging. The review text should focus to which of the strategies can be used in OA chondrocytes and the associated challenges.
- For instance, in section 4.2, extensive details have been provided on enhancing telomerase expression/activity; however, this strategy cannot be employed for OA. The section can be shortened to highlight the challenges of reversing telomere attrition in OA.
- Likewise, sections 4.1 & 4.3 do not discuss the therapeutic benefits of increasing genomic stability and epigenetic modifications for OA, respectively.
- Section should discuss the studies that have already been performed targeting aging-related pathways in OA, with details on in vivo, in vitro or ex vivo model systems, delivery systems for the therapeutic and percent reversal of phenotype.
Reviewer 3 Report
In the manuscript " Targeting cellular hallmarks of aging to reverse osteoarthritic
phenotype in chondrocytes", He et al. review current literature about the contribution of chondrocyte aging in osteoarthritic pathogenesis. Although several related reviews have already been published but in general, this review manuscript covers a broad scope of mechanisms of cellular senescence in OA development. However, there still some issues need to be solved:
- What's the different focus between this review and other reviews recently published on the related topic? For example, “Aging and osteoarthritis” PMID: 21709557, and “Cellular aging towards osteoarthritis” PMID: 28049007, and “Targeting aging for disease modification in osteoarthritis” PMID: 28957964, and “Ageing and the pathogenesis of osteoarthritis” PMID: 27192932.
- Gene therapy has attracted extensive attention in recent years, the author should summarize related studies in the treatment of OA. For example, lentivirus and genetic enhancement-based therapy.
- In the section of “Potential of targeting aging hallmarks to reverse OA chondrocytes”, the authors should add the defects/side effects of each potential strategy in targeting aging cells.
- Actually, exercise is also an important intervention strategy to delay aging. I am curious whether moderate exercise could alleviate OA.
- Line 210-212: lack of citation.
Round 2
Reviewer 1 Report
The authors made many changes according to the comments, which improved the quality and cleared the expression of the manuscript. I agree to publish this paper.
Reviewer 2 Report
The authors have addressed all of the comments and have modified the review appropriately.